# Phase-amplitude coupling supports phase coding in human ECoG

Andrew J Watrous[1]*, Lorena Deuker[1,2], Juergen Fell[1], Nikolai Axmacher[3,4]*

[1]Department of Epileptology, University of Bonn, Bonn, Germany; [2]Donders Institute for Brain, Cognition and Behaviour, Radboud University, Nijmegen, Netherlands; [3]German Center for Neurodegenerative Diseases, Bonn, Germany; [4]Department of Neuropsychology, Institute of Cognitive Neuroscience, Faculty of Psychology, Ruhr University Bochum, Bochum, Germany

**Abstract** Prior studies have shown that high-frequency activity (HFA) is modulated by the phase of low-frequency activity. This phenomenon of phase-amplitude coupling (PAC) is often interpreted as reflecting phase coding of neural representations, although evidence for this link is still lacking in humans. Here, we show that PAC indeed supports phase-dependent stimulus representations for categories. Six patients with medication-resistant epilepsy viewed images of faces, tools, houses, and scenes during simultaneous acquisition of intracranial recordings. Analyzing 167 electrodes, we observed PAC at 43% of electrodes. Further inspection of PAC revealed that category specific HFA modulations occurred at different phases and frequencies of the underlying low-frequency rhythm, permitting decoding of categorical information using the phase at which HFA events occurred. These results provide evidence for categorical phase-coded neural representations and are the first to show that PAC coincides with phase-dependent coding in the human brain.

*For correspondence:
andrew.j.watrous@gmail.com
(AJW); nikolai.axmacher@ruhr-uni-bochum.de (NA)

Competing interests: The authors declare that no competing interests exist.

## Introduction

Perceptual representations of the environment are critical to an animal's survival and are believed to occur through coactivated neuronal groups known as cell assemblies. Human neuronal firing (*Ekstrom et al., 2007*; *Kraskov et al., 2007*; *Chan et al., 2011*; *Rey et al., 2014*) and increases in high-frequency activity (HFA) in the gamma range (above 30 Hz; *Jacobs and Kahana, 2009*; *Jacobs et al., 2012*; *van Gerven et al., 2013*) carry information about perceptual and mnemonic representations. Several recent studies have shown that these two signals are positively correlated (*Ray et al., 2008*; *Manning et al., 2009*; *Whittingstall and Logothetis, 2009*; *Miller et al., 2014*; *Rey et al., 2014*; *Burke et al., 2015*) and are each modulated by the phase of low frequency oscillations (LFO) (*O'Keefe and Recce, 1993*; *Bragin et al., 1995*; *Skaggs et al., 1996*; *Canolty et al., 2006*; *Jacobs et al., 2007*; *Tort et al., 2009*; *Axmacher et al., 2010*; *Rutishauser et al., 2010*; *McGinn and Valiante, 2014*). This modulation is detectable as phase-amplitude coupling (PAC) of gamma amplitude to LFO phase (*Buzsaki, 2010*; *Miller et al., 2014*; *Aru et al., 2015*).

Together, these findings have motivated models positing that LFO phase may organize cell assemblies (*Kayser et al., 2012*; *Lisman and Jensen, 2013*; *Jensen et al., 2014*; *Watrous et al., 2015*), a form of phase coding (*O'Keefe and Recce, 1993*). Supporting this view, LFO phase can be used to decode behaviorally relevant information (*Belitski et al., 2008, 2010*; *Fell et al., 2008*; *Schyns et al., 2011*; *Lopour et al., 2013*; *Ng et al., 2013*) and phase coded neural activity has been demonstrated in rodents (*O'Keefe and Recce, 1993*; *Skaggs et al., 1996*) and monkeys (*Kayser et al., 2009*; *Siegel et al., 2009*). Although the PAC observed in humans (*Canolty et al., 2006*; *Axmacher et al., 2010*) has been thought to reflect phase-coding, this assumption has yet to be validated because prior studies have not investigated the relation between PAC and decoding from LFO phases.

**eLife digest** Electrocorticography, or ECoG, is a technique that is used to record the electrical activity of the brain via electrodes placed inside the skull. This electrical activity repeatedly rises and falls, and can therefore be represented as a series of waves. All waves have three basic properties: amplitude, frequency and phase. Amplitude describes the height of a wave's peaks (and the depth of its troughs), and frequency defines how many waves are produced per second. The phase of a wave changes from 0° to 360° between two consecutive peaks of that wave and then repeats, similar to the phases of the moon.

Previous studies have shown that brain activity at different frequencies can interact. For instance, neural firing (when nerve impulses are sent from one neuron to the next) is related to 'high frequency activity'; and the amplitude of high frequency activity can be altered by the phase of other, lower frequency brain activity. It has been suggested that this phenomenon, called 'phase-amplitude coupling', might be one way that the brain uses to represent information. This 'phase coding' hypothesis has been demonstrated in rodents but is largely untested in humans.

Now, Watrous et al. have explored this hypothesis in epilepsy patients who had ECoG electrodes implanted in their brains for a diagnostic procedure before surgery. These electrodes were used to record brain activity while the patients viewed images from four different categories (houses, scenes, tools and faces).

Watrous et al. found that phase-amplitude coupling occurred in over 40% of the recordings of brain activity. The analysis also revealed that the phase of the lower frequency activity at which the high frequency activity occurred was different for each of the four image categories. This provides support for the phase-coding hypothesis in humans. Furthermore, it suggests that not only how much neural firing occurs but also when (or specifically at what phase) it occurs is important for how the brain represents information. Future studies could now build on this analysis to see if phase-amplitude coupling also supports phase coding and neural representations in other thought processes, such as memory and navigation.

We have recently proposed that the frequency-specific phase of LFO coordinates neural firing to support neural representations (*Watrous and Ekstrom, 2014*; *Watrous et al., 2015*). Here, we tested this prediction, a form of the phase-coding hypothesis in humans, by examining the relation between PAC and neural representations for categories. We analyzed intracranial recordings from 167 electrodes in six patients with pharmaco-resistant epilepsy as they viewed pictures of houses, tools, scenes, and faces. First, we identified PAC on individual electrodes by using a recently developed metric which allows for the characterization of PAC across individual HFA events. On electrodes exhibiting PAC, we then assessed the distinctiveness of each category's phase-coded representation during periods with and without pronounced HFA. Our results suggest that during periods with pronounced HFA, categorical representations can be recovered based on the phase of low-frequency oscillations, supporting the idea of phase-coded neural representations in humans.

## Results

We analyzed data from a total of 167 intracranially-recorded EEG electrodes from six patients with pharmaco-resistant epilepsy as they viewed pictures of houses, faces, tools, and outdoor scenes (*Figure 1A*), testing whether these categories may be represented based on HFA activity at different phases of the LFO (*Figure 1B*). We first sought to identify electrodes exhibiting PAC and used a data-driven method which allows for the identification of predominant modulating and modulated frequencies (*Dvorak and Fenton, 2014*; see *Figure 1—figure supplement 1* for analysis schematic). *Figure 2A–D* shows the PAC modulation profile of an example electrode from the basal temporal lobe of patient 3. *Figure 2A* shows the magnitude of the modulatory signal relative to HFA events (time 0) at different frequencies in the HFA band. PAC is evident as red and blue vertical striping, with maximal modulation of activity at 84 Hz (*Figure 2B*; 'HFA event' marked by arrow in *Figure 2C*) occurring near the trough of the 2.5 Hz oscillation (see also *Figure 2D*). Notably, PAC was visible in the raw trace (*Figure 2C*), the modulatory signal showed rhythmicity (*Figure 2D*), and there was a clear peak in the power spectrum of both the raw signal and the modulatory signal (*Figure 2E*, see *Figure 2—figure supplement 1* for more examples).

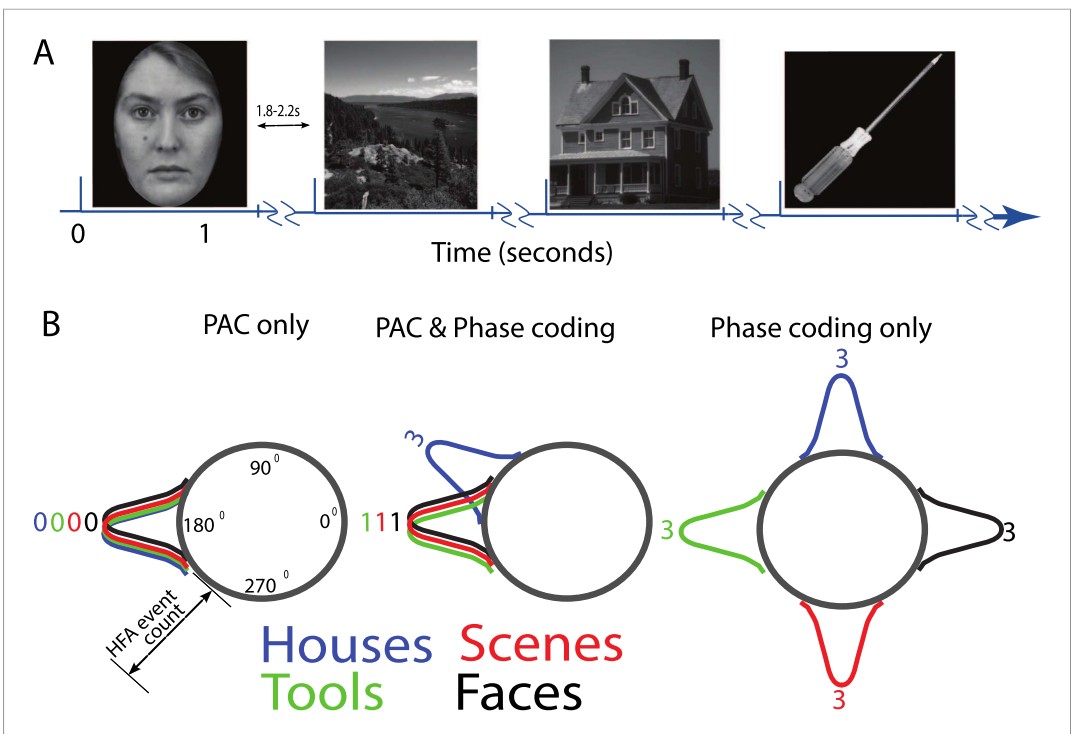

**Figure 1**. (**A**) Task structure and timing. Exemplar images are shown from each category. Each image was presented in pseudo-random order for one second with a jittered inter-stimulus interval. (**B**) Theoretical model of phase amplitude coupling (PAC) and phase coding, showing how each phenomenon could occur in isolation (left, right) or together (middle). Numbers above distributions indicate difference scores (DSs), the total number of categories one category differs from. High-frequency activity (HFA) may occur at specific phases but not differ between categories, leading to PAC without phase coding (left). Alternatively, HFA may be phase clustered across categories but still occur at different phases for some categories, leading to both PAC and phase coding (middle). In a third scenario (right), category-specific phase clustering could occur without any phase-clustering of HFA across categories, leading to phase coding without PAC.

The following figure supplement is available for figure 1:

**Figure supplement 1**. Schematic showing the calculation of oscillatory triggered coupling (OTC) and DS (panels **A** and **B**, respectively).

Next, we investigated the prevalence of PAC and HFA on each electrode. We found robust evidence for PAC, with at least 20% of electrodes in each patient showing significant PAC (n = 72/167 'PAC+' electrodes, see 'Materials and methods' for statistical assessment and inclusion criteria). On PAC+ electrodes, HFA was broadly distributed across trials and time points. Calculating the proportion of trials showing a period of significantly increased HFA (95th percentile, see 'Materials and methods' for 'HFA windows') on each PAC+ electrode and category, we found that HFA occurred throughout the period of stimulus presentation but increased ~150 ms after stimulus onset (*Figure 2F*). 66% of trials had at least one HFA window and this prevalence did not vary by category (*Figure 2—figure supplement 2*; one-way ANOVA, $F(3,284) = 0.6$, $p > 0.61$). These findings converge with prior studies demonstrating increased neural firing and HFA during stimulus presentation and demonstrate pronounced PAC in our paradigm (*Canolty et al., 2006*; *Mormann et al., 2008*; *Axmacher et al., 2010*; *Cichy et al., 2014*; *Rey et al., 2014*).

We then determined the frequencies and phases at which PAC is maximal on each PAC+ electrode. Slow-modulating ('$F_{phase}$') frequencies were significantly clustered in the delta band (0.5–4 Hz; *Figure 2G*) and HFA modulated frequencies ('$F_{amp}$') were significantly clustered around slow (~32 Hz) and fast (~110 Hz) gamma frequencies (*Figure 2H*, chi-square goodness of fit test across gamma frequencies, $p < 0.004$, $\chi^2(22) = 43.6$, Cohen's d = 0.77). Furthermore, we found that HFA was typically maximal near the trough of the oscillation (i.e., at 180°; *Figure 2I*; $p < 0.05$, Rayleigh test; see *Figure 2—figure supplement 1* for additional examples and modulation at other phases).

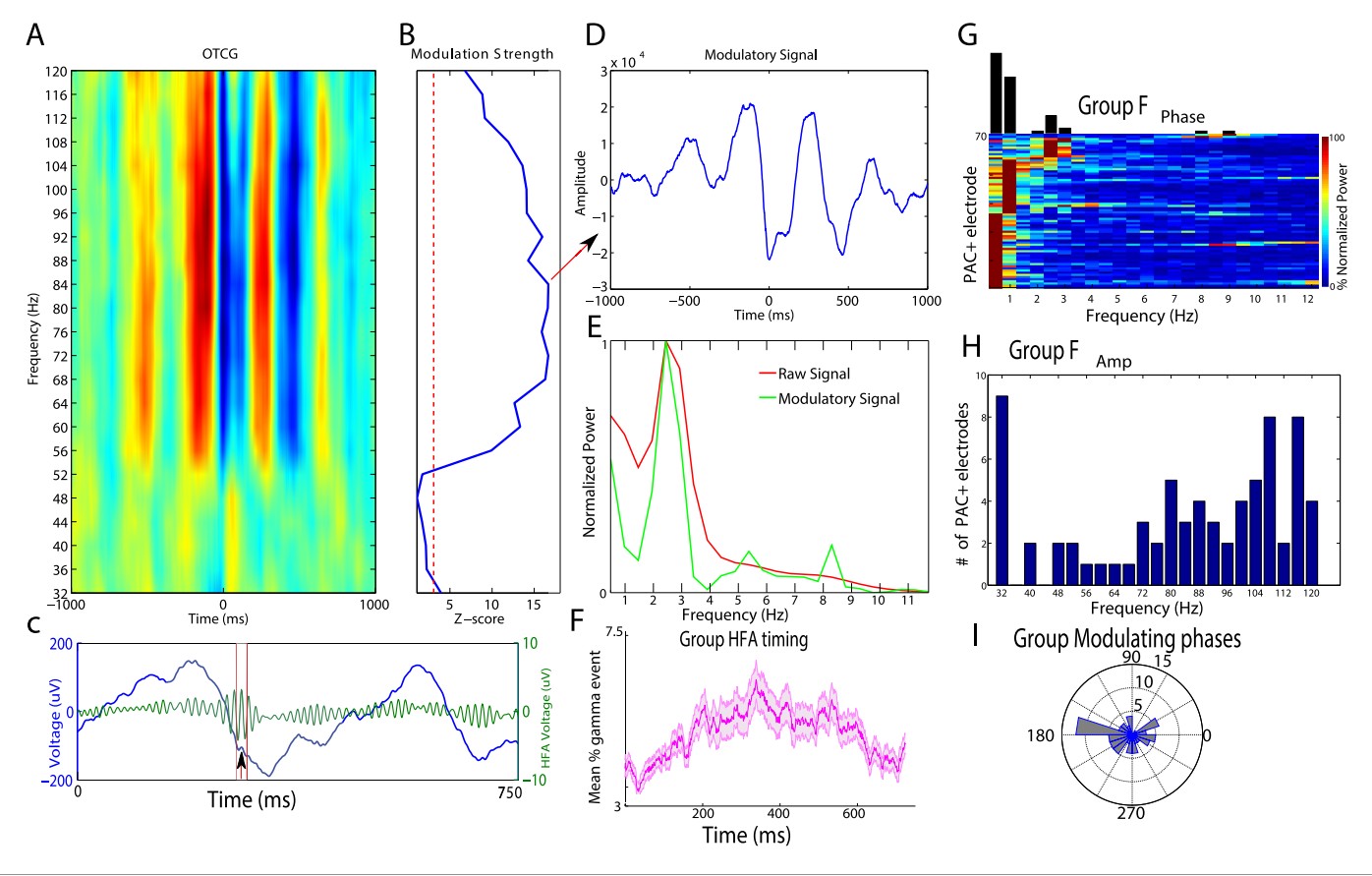

**Figure 2**. Phase amplitude coupling analysis. (**A–E**) Example of PAC using the OTC method described by *Dvorak and Fenton (2014)*. Data are from one electrode located in the left basal temporal lobe of patient #3. (**A**) Oscillatory-triggered comodulogram shows phase coupling above 50 Hz, evident as red and blue vertical striped regions. Time zero corresponds to the HFA event. (**B**) Z-scored modulation strength as a function of frequency relative to 100 surrogate shuffles at pseudo-HFA events (i.e., random time points). (**C**) Modulation of gamma amplitude (green) by the phase of a 2.5 Hz oscillation (blue) on an example trial. Time zero indicates image onset. Red shaded area and arrowhead indicate an HFA window and HFA event, respectively. Extracting the peak modulatory signal from **B** (84 Hz) reveals the phase (**D**, HFA events occur at the trough at time 0), strength (**D**, peak-to-trough height) and frequency (**E**; green) of the modulation. The red trace in (**E**) shows the average normalized power of the entire recording. (**F**) Group level analysis of HFA event timing. HFA events occurred throughout the stimulus presentation period but increased ~150 ms after stimulus onset. Magenta trace shows percentage of gamma events as a function of time, averaged across electrodes and categories. The timing of HFA events did not systematically differ by category (*Figure 2—figure supplement 2*). (**G**) Group level FFT data, defined at the peak of the modulation strength curve for each PAC+ electrode. Most PAC occurred around 1 Hz. Black bars are relative counts of electrodes with a peak at each frequency. (**H**) Distribution of modulated frequencies across electrodes. Electrodes were primarily modulated in the low and high gamma bands. (**I**) Preferred phases for modulation, clustered around the trough of the signal (180°).

The following figure supplements are available for figure 2:

**Figure supplement 1**. Additional examples of PAC from each patient, demonstrating frequency and phase-diversity of PAC.

**Figure supplement 2**. HFA time course for each category.

**Figure supplement 3**. Comparison of PAC results using the OTC and modulation index (MI).

We next tested if PAC occurs for all four categories, which would be necessary if PAC was related to the representation of categorical information. To this end, we tested each category separately for phase clustering of HFA events at the electrode-specific peak modulatory frequency ('$F_{MAX}$'). This analysis revealed significant clustering for all four categories on 87% (63/72) of PAC+ electrodes (Rayleigh test $p < 0.00004$, Bonferroni-corrected $p < 0.01$ for PAC+ electrodes and categories). Phase

clustering was observed in each patient and did not vary across categories at $F_{MAX}$ (one-way ANOVA on resultant vector lengths, F(3,284) = 0.14, p > 0.93). In sum, we found evidence for widespread PAC in each patient at several frequencies and phases of the LFO, similar to single neuron and field potential studies in monkeys (*Kayser et al., 2009*; *Siegel et al., 2009*) and humans (*Canolty et al., 2006*; *Jacobs et al., 2007*; *Axmacher et al., 2010*; *Maris et al., 2011*; *Jacobs et al., 2012*; *van der Meij et al., 2012*; *Voytek et al., 2015*).

## HFA occurs at different phases for different categories

Testing the phase-coding hypothesis, we asked if high frequency activity occurred during category-specific phases of the modulatory LFO. *Figure 3A* shows two traces from an example electrode which are color-coded by the instantaneous phase at $F_{max}$. HFA windows (boxes color coded by 1 Hz phase) occurred during different modulatory phases depending on stimulus category. On this electrode, phases extracted during HFA windows were clustered for each category to different phases, resulting in category-specific phase-clustering (*Figure 3B*). Similar findings were observed in other patients (*Figure 3C*), and appeared distinct from representations using power or phase (*Figure 3—figure supplements 1, 2*).

These findings imply that representations might occur by the category-specific phase at which HFA events occur. In order to further quantify this effect, we developed a simple metric, the difference score ('DS'), which allowed us to identify the distinctiveness of each category's phase distribution during HFA windows. We applied this metric to the subset of 63 PAC+ electrodes showing significant phase-clustered HFA for each category. This was necessary in order to exclude spurious phase differences between categories occurring in the absence of phase clustering. Across all patients, 78% (49/63) of PAC+ electrodes showed a unique phase-clustering profile for one category compared with each other category (e.g., *Figure 1B*; DS = 3 for at least one category, p < $10^{-9}$, Watson Williams test, Bonferroni corrected across comparisons). This pattern was consistent both within and across patients, with at least 15% of electrodes in each patient showing these effects (*Figure 3D*).

We next calculated the average phase difference between categories, expecting this measure to increase with increasing DS. Indeed, categories with larger DSs exhibited larger phase differences with other categories such that maximally distinct representations were 35° phase offset from all other categories (*Figure 3E*).

As described above, PAC was most likely to occur at the oscillatory trough (*Figure 2I* and *Figure 2—figure supplement 1*). Nonetheless, on individual electrodes or for individual categories, HFA could occur at different phases. In fact, across electrodes, phase-coding was equally likely to occur at all phases and for all categories; phase-coded categories were not clustered at particular phases at any level of DS (Rayleigh test, all p > 0.19; *Figure 3F*) and phase-coding was equally likely for each category ($\chi^2(3)$ = 1.6, p = 0.64). Thus, a large proportion of PAC+ electrodes also show category-specific phase clustering of HFA events to different phases (*Video 1*), suggesting that PAC is related to phase-coding (*Figure 1B*, middle).

## Decoding category identity from HFA event phases

To link these findings more directly to neural coding, we used pattern classification to determine if the phase at which HFA events occur is sufficient to recover categorical information (see 'Materials and methods'). As expected from the analysis using DS, 42 (25% of all) electrodes showed significant decoding accuracy (using LFO phases during HFA windows as features) compared to category label shuffled surrogates and this proportion was significantly higher than would be expected by chance (p < $10^{-10}$, binomial test, chance level: 8.3 electrodes, Cohen's d = 0.6). Next, we assessed whether phase-coding of categorical information indeed depended on HFA, as would be expected if PAC supports phase-coding. We compared decoding accuracy during HFA events to decoding accuracy during randomly selected surrogate events. 19 (11%) electrodes showed significantly higher decoding accuracy during HFA events as compared to random event surrogates, and this proportion was significantly higher than would be expected by chance (p < 0.0003, binomial test, chance = 8.3 electrodes, Cohen's d = 0.22). Moreover, 17 (10%) electrodes showed significant enhancements of decoding accuracy during HFA events relative to both label and event shuffled surrogates, with at least two electrodes in each patient showing this pattern. This proportion of electrodes far exceeded that expected by chance (p < $10^{-10}$, binomial test, chance = 0.41 electrodes, Cohen's d = 0.46).

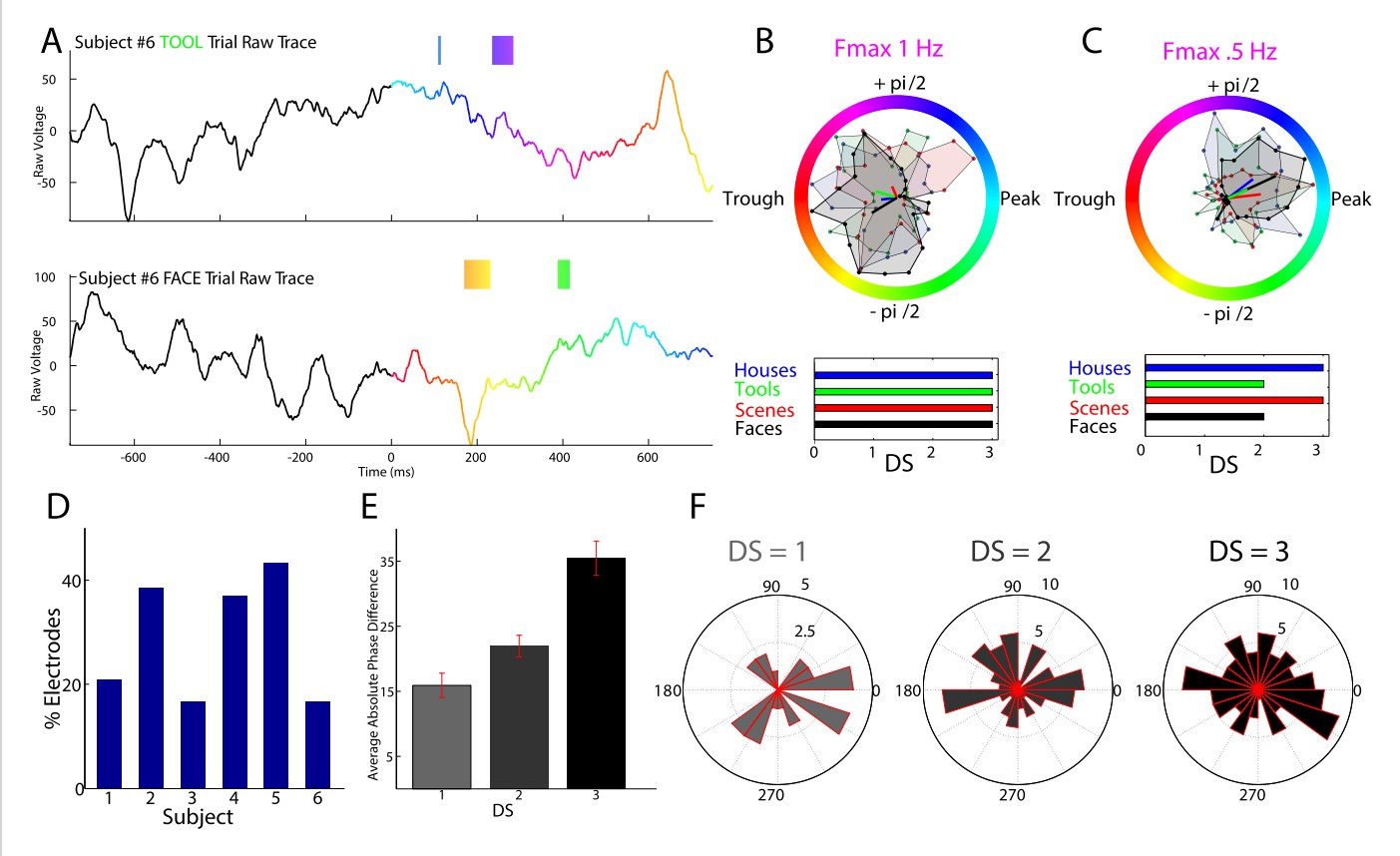

**Figure 3**. HFA occurs at category-specific low-frequency phases. (**A**) Two example trials from patient #6 demonstrating that HFA windows occur at different phases for different categories. The signal is color-coded by the phase of 1 Hz oscillation only during the stimulus period. Times prior to stimulus period are shown in order to visualize the 1 Hz modulatory signal. HFA windows are indicated by the boxes, color-coded by the 1 Hz phase at which they occur. (**B**) Summary circular histograms and resultant vectors for this electrode. Categorical phase-clustering to different phases was prominent at $F_{max}$, allowing for the decoding of categorical information based on the phase at which HFA events occur. DSs are plotted for each category in the lower panel. (**C**) Another example, from a different patient (#4), showing phase-clustered HFA windows for different categories (upper) along with DSs (lower). (**D**) Proportion of electrodes in each patient showing category specific phase-clustered HFA. (**E**) Average absolute phase difference across categories and electrodes for increasingly distinct phase representations (PRs). (**F**) Circular distribution of phases for each level of DS, pooled over electrodes and categories. Phase coded representations were equally likely to occur at each phase.

The following figure supplements are available for figure 3:

**Figure supplement 1**. Decoding categorical information using delta power, phase, or HFA power on example electrode shown in *Figure 3A–B*.

**Figure supplement 2**. Decoding categorical information using delta power, phase, or HFA power on example electrode shown in *Figure 3C*.

**Figure supplement 3**. Category-specific phase locking analysis.

These findings complement the above results using DS and indicate that the phase at which HFA events occur carries sufficient information to decode image category, suggesting such information may be a relevant component of the neural code.

We performed several control analyses to rule out alternative explanations. First, if slow oscillatory phase relates to category-specific representations, we expect phase-locking across trials to different categories. We observed significant phase locking on many electrodes to specific categories (*Figure 3—figure supplement 3*, Rayleigh test, p < 0.000001), similar to previous studies which have identified phase-locked activity (e.g., *Fell et al., 2008*). Second, we excluded the possibility that our PAC+ or phase-clustering inclusion criteria biased our findings by computing a composite measure of phase representation (PR) on each electrode (see 'Supplement results'). This analysis again revealed

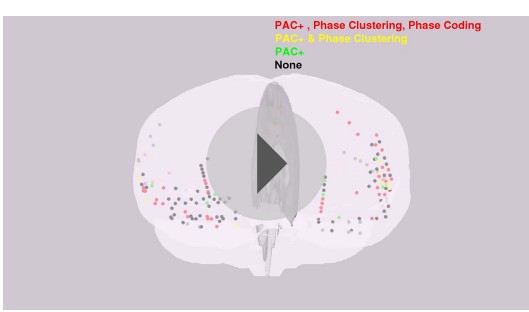

**Video 1.** Significant electrodes rendered onto a glass brain. Each point represents an electrode, and each color represents different effects. Black electrodes (n = 95) did not show significant phase-amplitude coupling (PAC). Green electrodes (n = 9) only showed significant PAC. Yellow electrodes (n = 14) showed significant PAC and phase-clustering of HFA for all 4 categories. Red electrodes (n = 49) showed significant PAC, phase-clustering for all 4 categories, and phase-coding of high-frequency activity (i.e., difference score of 3 for at least one category).

that phase coding is largest on PAC+ electrodes and is enhanced during HFA windows. Third, for comparison with prior PAC methods, we recomputed PAC using the modulation index (MI, *Tort et al., 2009*) in different low-frequency bands, again finding PAC that was most prevalent in the delta band (*Figure 2—figure supplement 3*).

Lastly, several models predict that neural processes forming representations will show frequency-specificity (*Siegel et al., 2012*; *Watrous and Ekstrom, 2014*; *Womelsdorf et al., 2014*). We therefore recalculated phase-clustering and DS at the minimum modulatory frequency ($F_{MIN}$; see 'Materials and methods' and *Figure 4—figure supplement 1* for individual subject values) using the same criteria detailed above. As one would expect, on PAC+ electrodes, phase clustering was larger at $F_{MAX}$ compared to at $F_{MIN}$, both on individual electrodes (*Figure 4A, B*) and at the group level (*Figure 4C*; paired t-test on resultant vector lengths, $t(287) = 8$, $p < 10^{-10}$, Cohen's d = 0.32). Moreover, only 20% (15/72) of PAC+ electrodes showed significant phase-clustering at $F_{MIN}$ for all 4 categories and only 1 electrode showed category-selective phase-clustering of HFA events. Given that the phase of slower frequencies varies less over time and that we primarily identified $F_{max}$ at slow frequencies, this result might be biased towards finding enhanced phase clustering at $F_{max}$. We therefore recalculated phase clustering across the full range of frequencies (1–12 Hz, 0.1 Hz steps). Again, we found enhanced phase-clustering around 0.5 and 1 Hz (*Figure 4—figure supplement 2*), but not at adjacent frequencies as would be expected from this alternative account. Taken together, these results support the conclusion that HFA at distinct phases and frequencies reflect representations for different categories.

## Discussion

We tested the hypothesis that PAC reflects a phase-coding mechanism, measuring both PAC and categorical PR in intracranial recordings from six patients who viewed pictures from different categories. Our analyses show that on a large subset of electrodes showing PAC, the frequency-specific phase at which HFA occurs varies with categorical information. Therefore, to the extent that HFA reflects increases in local neuronal activity (*Crone et al., 1998*; *Miller et al., 2014*), our results suggest that neural representations for categories might occur by the phase at which neurons fire. These findings thus provide a novel link between PAC and phase-coded neural representations in humans.

Critically, although PAC and phase-coded representations share some attributes, such as phase-clustering of activity, they are not necessarily identical processes. High frequency activity could occur at particular phases of LFOs, as reflected by PAC, but these phases may not vary with stimulus category (*Figure 1B*, left). In other words, there could be PAC without phase coding. This is in fact the null hypothesis we have tested and would manifest as PAC with DSs of zero. On the other hand, categorical information may be represented by specific low-frequency phases independent of HFA, leading to DS without phase-clustering across HFA events (PAC; *Figure 1B*, right). We did not find a complete overlap between PAC+ and phase-coding electrodes, indicating that each can occur in isolation, but instead found a compromise between these extremes (*Figure 1B*, middle). These results suggest that PAC in many cases reflects phase coding because of the significant overlap between the two phenomena (*Video 1*).

Phase-coding, in the form of phase-modulated neuronal firing, has been identified in rodents, monkeys, and humans (*O'Keefe and Recce, 1993*; *Skaggs et al., 1996*; *Jacobs et al., 2007*; *Kayser et al., 2009*; *Rutishauser et al., 2010*). Although the mechanisms which guide such a neuronal phase preference remain poorly understood, previous studies have found enhanced PAC during learning and memory tasks (Tort et al., 2008; *Axmacher et al., 2010*; *Kendrick et al., 2011*; *Friese et al., 2013*;

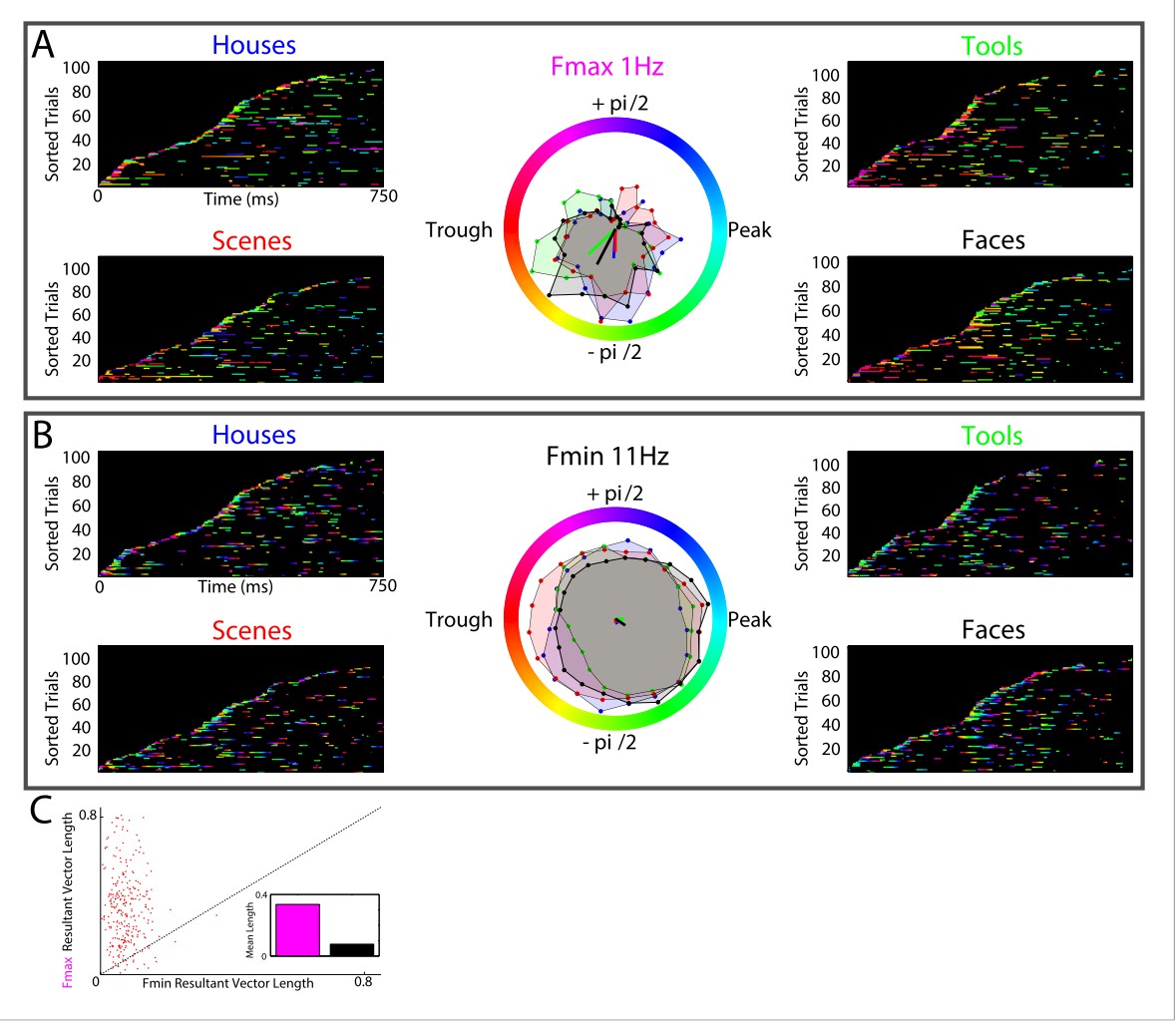

**Figure 4**. HFA clusters to specific phases and frequencies for different categories. (**A**) Example electrode showing phase clustering at the maximum modulatory signal ($F_{max}$; frequency with maximum power in the FFT, see *Figure 2E*) but not at the minimum modulatory signal (panel **B**; $F_{min}$; frequency with minimum power in the FFT). HFA events are marked in color as the phase of the oscillation at the respective frequencies. (**C**) At the group level, phase clustering was more prominent at the maximum frequency ($F_{max}$; maroon) compared to the minimum frequency ($F_{min}$; black) across categories and PAC+ electrodes.

The following figure supplements are available for figure 4:

**Figure supplement 1**. $F_{max}$ and $F_{min}$ values by subject.

**Figure supplement 2**. Phase-locking analysis across all frequencies from 0.1–12 Hz.

*Lega et al., 2014*). Our findings provide a potentially unifying account of these observations, suggesting that PAC may be promoting the formation of phase-coded neural assemblies (*Canolty and Knight, 2010*; *Watrous and Ekstrom, 2014*; *Watrous et al., 2015*). Follow-up studies will need to test this account of PAC as it relates to other putative roles for PAC (*Canolty and Knight, 2010*; *Voytek et al., 2015*).

While epilepsy is marked by increased synchronized neuronal activity which could potentially manifest as HFA or PAC, we believe several factors weigh against this interpretation. First, we only analyzed electrodes overlying putatively healthy tissue, typically from the hemisphere contralateral to the epileptic focus, as assessed by our clinical team. Electrodes showing epileptic spiking were systematically removed from our analysis and all analyzed trials were visually inspected for artifacts

related to epilepsy. Next, the PAC metric allows for assessment of the modulatory signal. Visual inspection of these signals did not reveal a similarity to epileptic spikes (*Figure 2—figure supplement 1*). Finally, it seems unlikely that epileptic activity at different phases would systematically differ by category. Similar reasoning excludes saccade-related artifacts (*Yuval-Greenberg et al., 2008*; *Kovatch et al., 2011*) as a parsimonious account of our results. We therefore conclude that similar findings would translate into healthy human populations.

Another caveat is that our results provide evidence for categorical phase-coding based on a restricted image set. This was necessary in the present study to maximize the chances of identifying category-selective responses while still ensuring that these responses were generalizable across a few exemplars. Follow-up studies should test the generalizability of these findings using more exemplars within a category and using other categories.

PAC has typically been investigated using pre-defined low and high-frequency filters which may optimize statistical power for detecting PAC but do not adequately deal with the time-resolved nature of cognition (*Aru et al., 2015*). Here, we leveraged a recent method which can identify PAC and subsequently test mechanistically interesting questions related to the modulation of HFA, such as its temporal profile and its dependence on phase, frequency, and behavioral requirements. Notably, this method may conservatively estimate PAC because it is based on transient increases in HFA, which do not necessarily occur in all cases of PAC. Our findings demonstrate that PAC and large HFA events can be identified and subsequently linked to categorically distinct representations. These results thus extend previous research which has decoded neural representations using either low or high frequency activity (*Jacobs and Kahana, 2009*; *Schyns et al., 2011*; *van Gerven et al., 2013*) and may provide new avenues for decoding the human representational system.

Intriguingly, phase-coding of categorical information extended beyond brain areas associated with higher-order vision. Thus, our findings of category-specificity do not appear to exclusively relate to perception but may also involve other more complex, and idiosyncratic, associations to these stimuli. Our findings are nonetheless in line with prior work (*Majima et al., 2014*; *Yaffe et al., 2014*; *Zhang et al., 2015*) which has found spatially-distributed content-specific representations.

We identified frequency-specific PRs in humans, consistent with a growing body of evidence implicating the relevance of frequency-specific oscillatory activity to human cognition (*Daitch et al., 2013*; *Watrous et al., 2013*; *Fontolan et al., 2014*; *Freudenburg et al., 2014*). These findings are therefore consistent with models implicating frequency-specific oscillations as central to higher-order cognition (*Siegel et al., 2012*; *Watrous and Ekstrom, 2014*; *Watrous et al., 2015*). It has recently been shown that the frequency of LFOs contributes to several neuronal properties such that relatively slower LFOs lead to decreased firing threshold and increased spike timing variability (*Cohen, 2014*). It is not immediately clear how this relates to our finding that PAC predominantly occurs with modulating frequencies in the delta band, particularly around 1 Hz. It is possible that our findings reflect the activation of assemblies during 'up' states which show a similar frequency profile (*Destexhe et al., 2007*) or that the applied method of identifying peaks in the spectrum biased our findings to find PAC at lower frequencies.

A third possibility, more likely in our view based on our results indicating multiple modulating frequencies per electrode (*Figure 2G*), is that the timing of our task (1 image per second with a jittered inter-stimulus interval) partially entrained slow oscillations forming an oscillatory hierarchy (*Lakatos et al., 2005*). Similarly, our results showing PAC at a variety of phases and frequencies (*Maris et al., 2011*; *van der Meij et al., 2012*), particularly near 32 Hz, might reflect a form of 'nested coupling' (*Kopell et al., 2010*) distinct from 'broadband' high gamma, which has been suggested to reflect population spiking (*Manning et al., 2009*; *Miller et al., 2014*). Future research may clarify this issue by comparing single neuron activity and HFA modulation during different perceptual tasks and by investigating their relation to hierarchical cross-frequency coupling.

To summarize, by identifying electrodes exhibiting both PAC and phase-coded neural representations for categories, our results employing direct brain recordings explicitly link phase-coupled neural activity to phase coding in humans.

## Materials and methods

### Epilepsy patients

Six right handed patients with pharmacoresistant epilepsy (mean age 31.8 years; 3 female) participated in the study. All patients were stereotactically implanted for diagnostic purposes. Medial temporal

depth electrodes (AD-Tech, Racine, WI, USA) with 10 cylindrical platinum-iridium contacts (diameter: 1.3 mm) were implanted in 1 patient, and 5 patients were implanted with subdural grid and strip electrodes with stainless-steel contacts (diameter: 4 mm) at temporal, frontal, and parietal sites (*Video 2*). Recordings were performed using a Stellate recording system (Stellate GmbH, Munich, Germany) at the Department of Epileptology, University of Bonn, Germany. The study was conducted according to the latest version of the Declaration of Helsinki and approved by the ethical committee of the medical faculty at the University of Bonn (approval identifier 280/08). All patients provided written informed consent to participate in the study and for the results to be published in a pseudonymized manner.

## Experimental design

Patients performed an object–location association task, though here we focus on neural representations for categories independent of memory encoding per se. Patients viewed greyscale images taken from four different categories (houses, tools, scenes, and faces) and each category had four unique stimuli, resulting in a stimulus set of 16 unique images. Example images from each category are shown in *Figure 1A*. Each image was presented 30 times in pseudo random order (total of 480 trials) and was followed by a white square in a fixed location. Patients were instructed to form object–location associations and to rate if they liked or disliked each image, thus ensuring that they were attending to each image presentation. Images were presented on a laptop placed in front of the patient. Each image was presented for 1 s, followed by the white square presented for 1 s, and finally a jittered inter-stimulus interval ranging from 1800–2200 ms. A fixation cross was presented between images.

## Recording and analyses

Intracranial EEG recordings (sampled at 1000 Hz) were referenced to linked mastoids and band-pass filtered (0.01 Hz [6 dB/octave] to 300 Hz [12 dB/octave]). Recordings from the hemisphere contralateral to the epileptogenic focus were analyzed. To boost our electrode sampling, an additional 32 electrodes from an ipsilateral left lateral temporal grid were included from patient 5 based on the physicians' report, which indicated a left hippocampal focus and no evidence of neocortical lesion based on an magnetic resonance imaging (MRI). Signals from this grid were carefully visually inspected for artifacts and did not show increased artifacts associated with epilepsy. Qualitatively similar results were observed when excluding these electrodes from the analysis, with the proportions of electrodes showing any reported effect changing by no more than 3%.

Electrode locations were determined by post-implantation MRI such that electrodes were mapped by co-registering pre- and post-implantation MRIs, normalizing the pre-implantation MRI and applying the normalization matrix to the post-implantation MRI. The anatomical locations of contacts were then identified by comparison with standardized anatomical atlases and using custom software (published at http://pylocator.thorstenkranz.de/). In total, 167 implanted electrode contacts were analyzed across all patients (*Video 2*).

Raw EEG signals were extracted from 750 ms before to 1500 ms after image onset. EEG trials were visually inspected for artifacts (e.g., epileptiform spikes), and trials with artifacts were excluded from further analysis (15% of all trials on average). Trial epochs were then concatenated for subsequent analysis described below. We analyzed an average of 103 trials per category and subject and there were no differences in total number of trials analyzed across categories (F(3,20) = 0.38, p = 0.76).

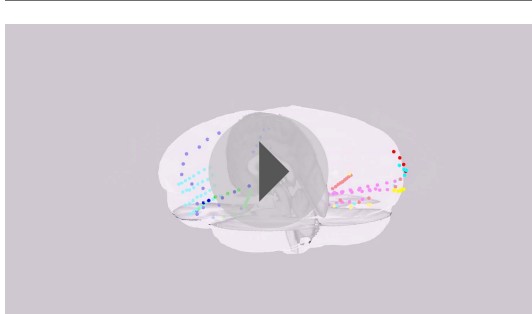

**Video 2.** Electrode locations for each patient, rendered onto a glass brain of the average MNI template. Each point represents an electrode, and each color represents a different patient. Electrodes were primarily located in the temporal lobe.

## Oscillatory triggered coupling (OTC) analysis

PAC was detected using the methods described by *Dvorak and Fenton (2014)*. This method is conceptually similar to an event-locked analysis around periods of enhanced HFA. All analyses

were conducted using standard routines in EEGLab (*Delorme and Makeig, 2004*) and Matlab based on previously published algorithms (*Rizzuto et al., 2006*; *Berens, 2009*; *Tort et al., 2009*; *Dvorak and Fenton, 2014*). In brief, the power and phase of the signal on each electrode was computed in the low frequency ($F_{phase}$, 0.5–12 Hz, 0.5 Hz steps) and gamma ($F_{amp}$, center frequencies at 32–120 Hz, 4 Hz steps) bands using Morlet wavelet convolution with 7 cycles. At each center HFA frequency, the time course of power values was z-scored and time periods exceeding the 95th percentile of these values were identified (we refer to these as 'HFA windows', red shaded area in *Figure 2C*). The time point of the largest power value within each window was identified and taken as the time-locking 'HFA event' for OTC analyses (arrow, *Figure 2C*). Two-second segments (1 s before to 1 s after each HFA event) of the raw signal were extracted around these timestamps and raw signal segments were summed at each time point across segments, resulting in the modulatory signal at each center HFA frequency.

The strength of modulation was determined based on the peak to trough height of the modulatory signal. Surrogate modulatory signals (n = 100) were constructed at each modulated frequency based on choosing an equal number of pseudo-HFA events at random timestamps and repeating the above procedure. Surrogate modulation strengths were extracted and used to z-score the observed modulation strength. PAC+ electrodes were identified as electrodes (1) with a modulation strength z-score >4.35 for at least one gamma frequency and (2) with a clear peak in the power spectrum of the raw signal at $F_{phase}$ (*Aru et al., 2015*). This z-score threshold was calculated by identifying the z value equivalent to a Bonferroni corrected (across 23 gamma frequencies and 167 electrodes) alpha threshold of $p < 0.05$ and corresponded to $p < 0.00005$. Peaks were identified by normalizing the power spectrum of both the raw and modulatory signal to their respective maxima and ensuring that both normalized signals were maximal (i.e., 1) at the same frequency.

We identified the peak HFA modulation frequency as the frequency with the largest z-score and extracted the modulatory signal (see *Figure 2B*). The phase and frequency content of the modulatory signal was determined using a Hilbert transform and fast Fourier transform, respectively. The modulatory signal was mean-centered prior to FFT in order to remove DC components. The maximum ('$F_{MAX}$') and minimum ('$F_{MIN}$') of this FFT output indicate the strongest and weakest slow-modulating frequencies in the 0.5–12 Hz band, respectively.

## DS and phase clustering calculation

Phase comparisons were conducted using a Watson Williams test following *Rizzuto et al (2006)* and using code taken from these eegtoolbox available at (http://memory.psych.upenn.edu/Software). Statistical testing was performed between the phases extracted during HFA windows for all pairs of conditions (4 categories; 6 total category pairs). DSs were computed for each category as the total number of significant differences ($p < 0.001$) between the phase distribution for one category and the remaining categories and thus ranged from 0 (no difference to any other category) to 3 (significant difference to all other categories). Phase clustering scores were defined as the resultant vector length for each category's phase distribution.

## Pattern classification analysis

We used a pattern classification approach for comparison with our DS metric, classifying image category based on the phase at which HFA events occur. To this end, we trained support vector machines using a linear kernel and fivefold cross-validation. Phase values at $F_{max}$ were extracted at moments in time when HFA events occurred during image presentation and were used as input features for the classifier. Similar to previous approaches (*Lopour et al., 2013*; *Majima et al., 2014*), the sine and cosine of the phase values were used as input features for phase. Classifiers were run separately on each electrode and the classifier output was a prediction of the category label for each HFA event. Classification accuracy was defined as the average proportion of correctly classified HFA events across folds.

Chance classification performance varies across electrodes because we classified the category label associated with each HFA event and the number of HFA events per category varied across electrodes. We thus opted to report significance based on permutation testing which accounts for the varying chance level across electrodes and assessed the significance of classification using two separate analyses which both utilized permutation testing. First, we randomized the category labels associated with HFA events and assessed classification accuracy. Second, we used random time points (also corresponding to random phases) as surrogate HFA events and assessed classification accuracy. Each type of permutation test was performed 50 times, resulting in a distribution of pseudo classification

accuracy values for each test. Observed classification accuracies at or above the 95th percentile of each of these distributions were deemed significant. Thus, we fixed the type 1 error rate for each test at 5% and, assuming independence between tests, we would therefore expect $0.0025 \times 167 = 0.42$ electrodes to show significance by chance for both permutation tests.

## Supplement information

### Methods: control analyses and PR scoring

Conducting an alternative analysis, we aimed to determine if phase coding was more prominent during HFA windows. To this end, we created a composite measure of PR on each electrode. Direct comparison of the observed DS value to a surrogate distribution is problematic because DS assumes values between 0–3. Therefore, in the case of an observed DS of 3, it is impossible to identify any surrogates larger than our observation. We thus created a composite and continuous measure of PR by multiplying DS and phase clustering values on each category–electrode pair.

Permutation tests were then performed by shuffling the temporal position of HFA windows. This method is similar to a method used previously (*Axmacher et al., 2008*). Specifically, we randomly reordered the positions of HFA and non-HFA windows and then recalculated DS, phase-clustering, and PR values for each category. Notably, this method maintains the distribution of HFA and non-HFA window durations while shuffling these windows relative to the phase series. Surrogate PR values were calculated 200 times per electrode and the observed PR value was compared against the 95th percentile of this surrogate distribution. PR values were also extracted during non-HFA windows as a second control condition.

### Supplement results

We computed a composite measure of PR by weighting each category's DS by its phase-clustering value. This measure combines two intuitive features of phase-coding, namely that information is represented at different phases and that activity is concentrated at these phases (captured by DS and phase-clustering values, respectively). Testing the specificity of HFA windows for phase coding, we found that 56% (94/167) of electrodes showed PR that was larger during HFA windows compared to both time-shifted surrogates and non-HFA windows for at least one category. Moreover, PR values were significantly larger on PAC+ electrodes compared to electrodes without significant PAC (two-sample t-test, $p < 0.000005$). Thus, we find that phase coding is largest on PAC+ electrodes and is enhanced during HFA windows.

We calculated the MI (*Tort et al., 2009*) by binning HFA (51–200 Hz) amplitude according to phase in either the delta (0.5–4 Hz), theta (4–8 Hz), alpha (8–13 Hz) or the entire low frequency (0.5–12 Hz) bands. Following surrogate control analyses, in which we randomly shuffled gamma values 500 times prior to calculating MI, we observed significant PAC in each band relative to shuffled gamma MI values. Consistent with our primary results, PAC was most prominent in the delta band using these methods. Furthermore, the magnitude of gamma band activity was maximal at a similar phase of delta oscillations (180°, oscillatory trough) as when assessed using the OTC method. Based on the MI based metric, we found that 102/167 electrodes exhibited significant PAC with delta band phase. These results are shown in *Figure 2—figure supplement 3*.

## Acknowledgements

NA was supported by funding from an Emmy Noether grant by the DFG (AX82/2). NA and JF received funding via SFB 1089. We thank Hui Zhang and Marcin Leszczynski for comments on an initial manuscript version.

## Additional information

### Funding

| Funder | Grant reference | Author |
| --- | --- | --- |
| Deutsche Forschungsgemeinschaft | DFG AX82/2 | Nikolai Axmacher |
| Deutsche Forschungsgemeinschaft | SFB 1089 | Juergen Fell, Nikolai Axmacher |

The funder had no role in study design, data collection and interpretation, or the decision to submit the work for publication.

## Author contributions

AJW, Conception and design, Analysis and interpretation of data, Drafting or revising the article, Contributed unpublished essential data or reagents; LD, Conception and design, Acquisition of data, Analysis and interpretation of data, Contributed unpublished essential data or reagents; JF, Conception and design, Analysis and interpretation of data, Drafting or revising the article; NA, Conception and design, Analysis and interpretation of data, Drafting or revising the article, Contributed unpublished essential data or reagents

## Ethics

Human subjects: The study was conducted according to the latest version of the Declaration of Helsinki and approved by the local ethics committee, and all patients provided written informed consent.

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
