## [Decision Letter]

Thank you for submitting your work entitled “Phase-amplitude coupling supports phase coding in human ECoG” for peer review at *eLife*. Your submission has been favorably evaluated by Timothy Behrens (Senior Editor), a Reviewing Editor, and three reviewers.

The reviewers have discussed their reviews with one another and the Reviewing Editor has drafted this decision to help you prepare a revised submission.

All of the reviewers felt the paper was strong, but there were several concerns that need to be addressed:

Reviewer #1:

Watrous et al. tackle the important question of whether phase coding is a mechanism for information encoding in the human brain. They test whether cross-frequency coupling (CFC) subserves phase dependent coding of distinct visual categories. The study is highly relevant for the field of systems neuroscience.

1) The authors may be unintentionally inflating their findings by defining PAC by power increases in HFA through the oscillation triggered coupling (OTC) procedure. PAC does not require a change in power, but a change in the distribution of power relative to the phase of the low-frequency oscillation. Evoked activity might distort CFC results. Did the authors attempt to study evoked and non-stimulus locked activity separately?

2) OTC needs more explanation in this paper. Given the major claims of the paper, the authors should compare the OTC to more established PAC methods, so that the reader can better assess the results. The findings may be inflated by the chosen metric.

3) There are reports (Mathewson, 2009; Schyns, 2011) that indicate that phase coding only becomes evident in stages of high power. It is necessary to test the phase vs. power coding and then assess phase coding separately for high and low power trials. This is important, since the decoding results imply that HFA alone might explain some of the findings. F_MAX_ and F_MIN_ values per subject should be reported. In order to establish PAC as a putative mechanism, one needs to rule out that neither HFA nor low frequency phase/power alone can explain the findings.

4) Could the distribution of phase angles at stimulus onset exhibit a systematic bias with respect to the visual stream? To rule out entrainment to the rhythmic stream the authors could employ ITC or phase-locking to the sensory stream (e.g. Thut et al., 2011).

5) A major concern is the use of wavelets for extracting oscillatory phase. Wavelets constitute an acausal approach to determine the phase of a signal, since an underlying oscillation is assumed. This leads to frequency, but also phase smoothing, e.g. that a transient ERP contaminates the pre-stimulus phase estimates by backward ‘smearing’. The paper would benefit from a causal approach. Please consult Zoefel and Heil (2013, FIPSY).

6) The difference score is introduced in the second paragraph of the subsection “HFA occurs at different phases for different categories” and requires earlier explanation. A schematic for OTC and DS would be preferable.

7) Were the phase and power extracted from the same electrode? Was PAC calculated on epoched data? Both things could possibly lead to spurious findings.

8) Why did the authors only analyze the frequency spectrum up to 120 Hz? Often findings in HG in ECoG only start at 100 Hz and then extend up to 250 Hz.

9) Were eye movements recorded in any of the subjects? The authors argue that this is an unlikely source of artifactual data but eye-tracking data would be more assuring.

Reviewer #2:

A growing literature in systems neuroscience has observed a consistent relationship between the timing of spikes in local ensembles, relative to the phase position of local oscillatory field potentials. This putative phase-coding mechanism provides an interesting means of coordinating population coding, and makes clear predictions about temporal constraints on computation. To date, phase-coding has been studied in rodent models and non-human primates. In their submission, Watrous et al. seek to quantify a correlate of phase-coding in human cortex using direct intracranial recordings. As a means of quantifying phase-coding, the authors focus on phase-amplitude coupling (PAC) in the cortical surface potential, specifically low frequency phase modulation of high-frequency amplitude (HFA). This approach is supported by the correlation between HFA and local population spiking, and previous evidence of phase modulation of HFA. In this regard, HFA PAC reflects a macro-scale measure of population phase-coding.

The role of PAC in cognition is an active area of investigation and the author's attempts to link this phenomenon to phase-coding using human intracranial recordings is of great interest. At a general level, the relationship between phase-coding and local field potential PAC is unclear given the population level readout of spiking in the HFA. In their study, Watrous et al. attempt to identify unique PAC based phase-coding for different visual categories. The authors provide a clear motivation, and employ sophisticated analysis methods towards this endeavor.

1) Anatomy: While the authors provide evidence for PAC, and phase based coding of visual categories, their conclusions make no reference to functional neuroanatomy. Recording sites across subjects come from a wide variety of regions; this strikes me as a challenge when interpreting the specific coding of visual categories. The supplemental movie suggests a range of distant regions, not typically associated with higher-level vision, display some degree of phase-coding/representation of visual categories. This seems inconsistent with previous electrophysiology and functional imaging work (which is not cited or discussed in any detail). On a related note, the diversity of recording sites also makes the use of count statistics as somewhat arbitrary. Percentage of electrodes displaying an effect of interest can be equally meaningful for large or small percentages (e.g. low % for anatomically specific effect or high % for a trivial global effect). The authors should clarify why macro-scale phase-coding exists for visual categories across many cortical regions, rather than focused to the more classical regions of categorical selectivity.

2) Task/decoding: In their task, the authors present only four exemplars for each category, repeating each 33 times (if I understand correctly). While this may serve to aid mnemonic encoding, it does limit the claim of categorical decoding. Specifically, in developing a category decoder, the large number of stimulus repetitions limits insight, given the similarity between any training and test set. On a related note, I found the reporting of the basic decoding results unclear (is the decoder working above expected chance levels? >25%). Given the authors’ aims, it seems that a better use of the data would be quantifying the consistency of phase-coding metrics across repetitions of stimuli, as well as within/between class comparison. This approach would focus more on extracting single trial features and testing similarity across repeated trials (I note issues of repetition suppression come into play here). This approach of displaying consistency of stimulus phase-coding would provide more robust evidence for the authors’ claims.

Reviewer #3:

The manuscript by Watrous and colleagues is an interesting look at phase coding in the human cortex and medial temporal lobe. While the authors have a great deal of experience with ECoG analyses, including phase coding and PAC, and their manuscript is generally of interest, I have a number of questions and concerns.

Technical comments:

1) In Figure 2, Figure 2—figure supplement 1, the authors show “low gamma” PAC and “high gamma” PAC. This 32 Hz coupling mode seems striking, because it's likely that coupling extends even further below this range into the beta range. This low gamma has been argued to be distinct from more “broadband” high gamma (Kai Miller and Dora Hermes' work), which is correlated with population spiking (Mukamel, Science; Manning, J. Neurosci.), in contrast to low gamma, which is more oscillatory. Thus, the low gamma effect may be more a form of “nested” coupling as has been argued by Nancy Kopell.

2) There appears to be a disproportionate PAC effect at 0.5Hz and 1.0Hz, but with surprising specificity, and not between those two frequencies as seen in Figure 4—figure supplement 1. Why do the authors believe this occurs, and why do they believe their PAC effects are so restricted to this delta range, in contrast to what others have observed in ECoG?

3) How sensitive is detection of HFA event times to the filtering method?

4) With regards to electrode choice, the rationale for only using electrodes in the contralateral hemisphere is unclear. Why systematically reject an entire hemisphere (except for 1 subject, oddly) when you visually inspect channels for epileptic activity anyway? Additionally, what is the medical justification for implanting patients with electrodes in what is putatively a healthy hemisphere?

Statistics comments:

1) Watson-Williams test assumes a von Mises distribution. Is this true for the distributions studied here? If not, use the Wheeler-Watson test.

2) For the resampling statistics: the images were shown in groups of four, but the resampling seems to use random permutation. Resampling should be performed such that the labels for the “chunks” should be shuffled, but within these 4-trial chunks, the labels should be kept the same. This would control for any effect of this chunking.

3) Are there still significant differences between categories? How many electrodes have a category with DS=3?

4) It would be nice to also be given an estimate of effect size wherever a p-value is given.

5) For the SVM bootstrapping estimates, are the two bootstrapping experiments actually independent in order to support the expected false alarm rate of 0.42 electrodes?

General comments:

1) Are there spatial clusters among the electrodes that have phase coding for each of the different categories (c.f. Vidal et al, 2010)?

2) Please make all rose phase plots opaque as in Figure 3 so that we can see the phase distributions for each category.

3) For these phase plots, it would be nice to see the true number of high frequency activity events within each phase bin.

4) It is unclear how Figure 3—figure supplement 1 should be interpreted. For example, the primary effect in the paper is in the delta range, but this figure seems to show poor delta phase clustering. Why?

---

## [Author Response]

Reviewer #1:

1) The authors may be unintentionally inflating their findings by defining PAC by power increases in HFA through the oscillation triggered coupling (OTC) procedure. PAC does not require a change in power, but a change in the distribution of power relative to the phase of the low-frequency oscillation. Evoked activity might distort CFC results. Did the authors attempt to study evoked and non-stimulus locked activity separately?

As described in response to concern #2, we choose this relatively uncommon and novel metric of PAC because it is not biased by inter-trial phase locking of low-frequency activity. As the reviewer correctly points out, the OTC method is based on transient increases in HFA, which do not necessarily occur in all cases of PAC. We assessed PAC as both an increase in HFA and a phase-locking of HFA to slow oscillation phases. Given this additional constraint in our analysis, we believe we are conservatively estimating PAC because, as the reviewer notes, PAC is not necessarily associated with a change in power. This becomes apparent in our alternative metric (see response to concern #2). Regarding the issue of evoked activity, we observed HFA events distributed throughout the stimulus presentation period (Figure 2; Figure 3—figure supplement 1; Figure 3—figure supplement 1), which is inconsistent with the typical notion of stimulus evoked activity. We have added the following text to the manuscript: “Notably, this method may conservatively estimate PAC because it is based on transient increases in HFA, which do not necessarily occur in all cases of PAC”.

2) OTC needs more explanation in this paper. Given the major claims of the paper, the authors should compare the OTC to more established PAC methods, so that the reader can better assess the results. The findings may be inflated by the chosen metric.

We thank the reviewer for this suggestion and have conducted several analyses to this end. We calculated the Modulation index (MI: [56], PNAS) by binning HFA (51-200 Hz; see response to concern #8 for justification of this more extended high-frequency range) amplitude according to low-frequency phase in either the delta (.5-4 Hz), theta (4-8 Hz), alpha (8-13 Hz) or the entire low frequency (.5-12 Hz) bands. Following surrogate control analyses, we observed significant PAC in each band. Consistent with our primary results, PAC was most prominent in the delta band using these methods. Furthermore, the magnitude of gamma band activity was maximal at a similar phase of delta oscillations (180°, oscillatory trough) as when assessed using the OTC method. Based on the MI based metric, we found that 102/167 electrodes exhibited significant PAC with delta band phase. However, we found more electrodes showing significant PAC using the more traditional MI based metric compared to the OTC metric (72/167 PAC+ electrodes). We believe this is because, as the reviewer points out, MI does not require a change in power, whereas OTC is more conservative in requiring a change in power.

We opted for OTC rather than other PAC measures because we sought to interrogate the relation between PAC and phase-coding, which is thought to rely on precise timing and is calculated at individual time points. Any conventional time-resolved PAC metric such as MI (calculated across trials) is potentially biased by phase locking (and becomes impossible to calculate if inter-trial phase locking is perfect, because then no phase variance exists). On the other hand, we could not calculate PAC across time because of our relatively short trial periods (related to the reviewer’s concern regarding calculating PAC on epoched data). This reasoning motivated us to use the OTC method.

In addition, we have added a description of the analysis into the supplemental materials and have added the following text to the Results section to address this point: “Third, for comparison with prior PAC methods, we recomputed PAC using the Modulation Index (56) in different low-frequency bands, again finding PAC that was most prevalent in the delta band (Figure 2—figure supplement 3).”

*3) There are reports (Mathewson, 2009; Schyns, 2011) that indicate that phase coding only becomes evident in stages of high power. It is necessary to test the phase vs. power coding and then assess phase coding separately for high and low power trials. This is important, since the decoding results imply that HFA alone might explain some of the findings. F*_*MAX*_
*and F*_*MIN*_
*values per subject should be reported. In order to establish PAC as a putative mechanism, one needs to rule out that neither HFA nor low frequency phase/power alone can explain the findings.*

We fully agree with the reviewer, and in fact, an earlier version of this manuscript included such an analysis assessing categorical decoding using low frequency power/phase or HFA along with our difference score approach. We now include results from this analysis for the example electrodes in Figure 3. The left panel in Figure 3—figure supplement 1 shows the time-resolved and trial-averaged delta power/phase or HFA power values, along with difference scores, for each category. For comparison, the right panel shows HFA windows for each category (similar to Figure 4). Consistent with prior work (reviewed in [61]), these examples indicate that some categorical information is contained in each measure. In the electrode example in Figure 3—figure supplement 1, faces can be well distinguished from the other categories based on power information between around 300-700ms, based on low-frequency phase between around 100-150ms, and based on HFA between around 250-300ms. Critically, however, maximal phase and HFA differences scores (e.g. for faces) do not overlap and these enhanced DS periods do not neatly map onto HFA windows (compare to right panel). In this example, faces can be decoded via phase-clustered HFA (i.e. PAC) at many time points throughout the entire stimulus presentation period, while decoding based on low-frequency phase or power alone or based on the magnitude of HFA alone is only possible in restricted time periods. Moreover, unlike what we observe in our primary analysis based on PAC (Figure 3), categorical decoding for all categories is not possible with HFA alone (which in this example only allows distinguishing faces from non-faces, but not between any of the other categories).

Another example is shown in Figure 3—figure supplement 2 (same patient and electrode as in Figure 3). In this case, PAC-based decoding allows separating of both houses and scenes from all three respective other categories (PAC-based DS of 3 for houses and scenes in Figure 3). By contrast, unequivocal decoding by phase is only possible at specific time points for faces and tools (left panel in Figure 3—figure supplement 2), and decoding by HFA alone only allows separating faces from the three other categories at a short time window late in the trial.

These examples highlight a complex relationship between each measure and categorical representation, which certainly warrants further investigation. We believe a full characterization of these effects is beyond the scope of this manuscript. However, our examples clearly indicate that some categorical information can be recovered from low-frequency phase, power and HFA alone, but also demonstrate that additional information can be recovered based on PAC.

Finally, we now plot individual subject F_MAX_ and F_MIN_ values in Figure 4—figure supplement 1.

4) Could the distribution of phase angles at stimulus onset exhibit a systematic bias with respect to the visual stream? To rule out entrainment to the rhythmic stream the authors could employ ITC or phase-locking to the sensory stream (e.g. Thut et al., 2011).

We provided a preliminary evaluation of the issue regarding phase-locking in the former Figure 3—figure supplement 1 (now Figure 3—figure supplement 3). This figure shows the proportion of electrodes with significant phase-locking exclusively for each of the four categories as a function of time and frequency. We find evidence that there is some category-selective phase locking in our recordings. Importantly, our PAC results were strongest in the delta band using both OTC and MI methods ([56], PNAS) and thus do support some level of visual entrainment to stimulus presentation as described in the Discussion. However, overall only relatively small numbers of electrodes showed category specific phase-clustering (5 percent of all electrodes for all categories apart from tools, where 16 percent showed phase-clustering), while PAC-based decoding of each of these categories was possible in more than twice as many electrodes (12 to 17 percent of all electrodes had a difference score of 3 for each of the four categories). Again, this shows that PAC-based decoding cannot be explained only by phase-locking.

5) A major concern is the use of wavelets for extracting oscillatory phase. Wavelets constitute an acausal approach to determine the phase of a signal, since an underlying oscillation is assumed. This leads to frequency, but also phase smoothing, e.g. that a transient ERP contaminates the pre-stimulus phase estimates by backward ‘smearing’. The paper would benefit from a causal approach. Please consult Zoefel and Heil (2013, FIPSY).

We agree that extracting the phase of the signal using wavelets could potentially introduce phase-smoothing based on the assumption of an underlying oscillation but believe several considerations argue against such an effect confounding our results. First, visual comparison of raw signals and extracted phases (using wavelets) did not reveal any systematic shift in phase estimates (e.g. Figure 3). More importantly, we tested the assumption of an underlying oscillation directly by insuring that all PAC+ electrodes showed a peak in the power spectrum of the modulatory signal at the same frequency as the raw signal. Third, given that we are not exclusively looking at evoked activity, the potential of an ERP “backward smearing” our phase estimates is minimized, in theory. Finally, such phase smoothing might result in uniformly distributed (i.e. Von Mises) phases, which is the opposite of what we found when calculating phase-clustering (using wavelet-extracted phase estimates) on PAC+ electrodes. These considerations led us to use wavelets because they were used in the original implementation of OTC by Dvorak and Fenton.

6) The difference score is introduced in the second paragraph of the subsection “HFA occurs at different phases for different categories” and requires earlier explanation. A schematic for OTC and DS would be preferable.

Following up on this helpful comment, we have added a schematic flow chart for both the OTC method and DS as Figure 1—figure supplement 1.

7) Were the phase and power extracted from the same electrode? Was PAC calculated on epoched data? Both things could possibly lead to spurious findings.

Phase and power were extracted on the same electrode because we were interested in how categorical representation via phase might occur locally. Mathematically, phase and power are independent. Practically, however, they depend on the synchronization of neural assemblies nearby to the recording contact and thus are interdependent (e.g. high synchronization will result in a robust phase estimate and probably high inter-trial phase locking, but also in high power). In this study, it was no real option for us to use different electrodes because a main aim was to first assess PAC, which has primarily been characterized as a local phenomenon. Although some reports indicate that information might be represented between electrode sites (Canolty et al., 2010 PNAS; [40] Neuroimage), we believe this is the most straightforward way to understand phase coding. Furthermore, our new results on difference scores based on low-frequency power and low-frequency phase (see Figure 3—figure supplement 1 and Figure 3—figure supplement 2) show independent decoding from phase and power, speaking against the idea that these two metrics are strongly inter-related. PAC was calculated on concatenated epochs of artifact-rejected data using the OTC methods as described by [20]. Our analysis did not include HFA events detected near the concatenation border and thus we observed similar results when calculating OTC on unepoched or epoched but concatenated data (example taken from Figure 2—figure supplement 1 Patient #4, Results).

Author response image 1.**DOI:**
http://dx.doi.org/10.7554/eLife.07886.020

8) Why did the authors only analyze the frequency spectrum up to 120 Hz? Often findings in HG in ECoG only start at 100 Hz and then extend up to 250 Hz.

We opted to analyze the spectrum up to 120 Hz in order to avoid potential confounds associated with epilepsy, such as pathological “fast ripples” (above 150 Hz; Staba et al, 2007, Epilepsia; Menendez de la Prida et al, 2015, Journal of Clinical Neurophysiology). 120 Hz was also the upper frequency bound used by Dvorak and Fenton in the original paper describing OTC, and several other papers investigating PAC in humans have similar cutoff frequencies at or below 150 Hz ([10], Science; [37], Cerebral Cortex; [59], Nature Neuroscience). Given that the exact relation between normal/pathological HFA and cognition remains unclear, we thought it was most important to take a conservative approach on this matter though we do not dispute other studies which have found high gamma/HFA effects at higher frequencies.

We also note that we found similar results using the Modulation Index with the “full” gamma frequency range (50-200 Hz). Based on these reasons, we would suggest keeping the original frequency range. Nonetheless, we are willing to perform this additional analysis if the reviewer feels it is necessary.

9) Were eye movements recorded in any of the subjects? The authors argue that this is an unlikely source of artifactual data but eye-tracking data would be more assuring.

We instructed patients to keep their eyes on the fixation cross between stimuli in order to minimize any effects associated with eye movements. Unfortunately, electrooculogram/eye trackers are not well-tolerated by patients who already experience head discomfort associated with electrode implantation. Therefore, eye movement data was not systematically collected in this study, and thus eye movements are a potential confound which we cannot fully rule out (Kovach et al., 2011, Neuroimage). However, even when entertaining a scenario in which some of our HFA events were related to eye movements, it does not necessarily follow that: 1) these HFA events should show phase-locked activity, and 2) are also diagnostic of image category. Furthermore, eye movements occur predominantly at specific time points, in particular at around 200ms, where they may account for apparent induced gamma-band responses, while PAC in our study occurred widely distributed throughout the entire stimulus presentation phase (Figure 2—figure supplement 2).

Reviewer #2:

1) Anatomy: While the authors provide evidence for PAC, and phase based coding of visual categories, their conclusions make no reference to functional neuroanatomy. Recording sites across subjects come from a wide variety of regions; this strikes me as a challenge when interpreting the specific coding of visual categories. The supplemental movie suggests a range of distant regions, not typically associated with higher-level vision, display some degree of phase-coding/representation of visual categories. This seems inconsistent with previous electrophysiology and functional imaging work (which is not cited or discussed in any detail). On a related note, the diversity of recording sites also makes the use of count statistics as somewhat arbitrary. Percentage of electrodes displaying an effect of interest can be equally meaningful for large or small percentages (e.g. low % for anatomically specific effect or high % for a trivial global effect). The authors should clarify why macro-scale phase-coding exists for visual categories across many cortical regions, rather than focused to the more classical regions of categorical selectivity.

We thank the reviewer for this point and have added the following paragraph to the Discussion: “Intriguingly, phase-coding of categorical information extended beyond brain areas associated with higher-order vision. […] Our findings are nonetheless in line with prior work (Zhang et al., 2014; [40]; [65]) which has found spatially- distributed content-specific representations.”

2) Task/decoding: In their task, the authors present only four exemplars for each category, repeating each 33 times (if I understand correctly). While this may serve to aid mnemonic encoding, it does limit the claim of categorical decoding. Specifically, in developing a category decoder, the large number of stimulus repetitions limits insight, given the similarity between any training and test set.

We agree with the reviewer that the repetition of each exemplar (4 exemplars from 4 categories; 16 total images repeated 30 times each) limits the generalizability of our findings to other stimuli within these categories. Such compromises between generalizability and statistical sampling/power are a common problem in the field without any agreed upon solution. We opted for this design in order to maximize the chances of identifying category-selective responses while still ensuring that these responses were generalizable across a few exemplars. We have added the following text related to this point to the Discussion: “Another caveat is that our results provide evidence for categorical phase-coding based on a restricted image set. […] Follow-up studies should test the generalizability of these findings using more exemplars within a category and using other categories.”

On a related note, I found the reporting of the basic decoding results unclear (is the decoder working above expected chance levels? >25%).

We apologize for the lack of clarity in the reporting of the basic decoding results. Yes, all electrodes reported to be significant were at or above the 95th percentile of chance classification accuracies obtained using permutation testing. We classified the category label associated with each HFA event and the number of HFA events per category varied across electrodes. Therefore, chance classification performance varied across electrodes and was not fixed at 25%. We thus opted to report significance based on permutation testing which accounts for the varying chance level across electrodes. We have added to this effect in the Methods section: “Chance classification performance varies across electrodes because we classified the category label associated with each HFA event and the number of HFA events per category varied across electrodes. We thus opted to report significance based on permutation testing which accounts for the varying chance level across electrodes.”

Given the authors’ aims, it seems that a better use of the data would be quantifying the consistency of phase-coding metrics across repetitions of stimuli, as well as within/between class comparison. This approach would focus more on extracting single trial features and testing similarity across repeated trials (I note issues of repetition suppression come into play here). This approach of displaying consistency of stimulus phase-coding would provide more robust evidence for the authors’ claims.

We are not entirely clear what the reviewer is proposing here but feel that our analysis likely already incorporates these ideas. Our analyses were in fact based on the extraction of single-trial features. We insured that there was within class consistency for these features and then tested for how these might differ between categories. More specifically, we insured that the phases associated with HFA events (our single trial features) for each category were consistent by assessing phase-clustering locked to the onset of HFA events. We then compared these phase distributions between categories to test for phase-coding.

Reviewer #3:

*1) In*
Figure 2*,*
Figure 2—figure supplement 1*, the authors show “low gamma” PAC and “high gamma” PAC. This 32 Hz coupling mode seems striking, because it's likely that coupling extends even further below this range into the beta range. This low gamma has been argued to be distinct from more “broadband” high gamma (Kai Miller and Dora Hermes' work), which is correlated with population spiking (Mukamel, Science; Manning, J Neurosci), in contrast to low gamma, which is more oscillatory. Thus, the low gamma effect may be more a form of “nested” coupling as has been argued by Nancy Kopell.*

We thank the reviewer for this insightful comment. We agree that our results might imply several different “coupling modes”, perhaps with different forms of narrowband (low frequency) or broadband (higher frequency) gamma. We view this as a fruitful avenue for future study, and have added text to this effect in the Discussion: “Similarly, our results showing PAC at a variety of phases and frequencies (42; 57), particularly near 32 Hz, might reflect a form of ‘nested coupling’ (33) distinct from ‘broadband’ high gamma, which has been suggested to reflect population spiking (41; 45)”.

*2) There appears to be a disproportionate PAC effect at 0.5Hz and 1.0Hz, but with surprising specificity, and not between those two frequencies as seen in*
Figure 4—figure supplement 1*. Why do the authors believe this occurs, and why do they believe their PAC effects are so restricted to this delta range, in contrast to what others have observed in ECoG?*

We agree that this result is somewhat surprising and made several attempts to rule out trivial, signal processing based interpretations. We first wanted to make sure that the results weren’t associated with the 1/f nature of the signal and excluded this account based on the fact that there are distinct peaks at .5 and 1 Hz, rather than a smooth falloff with increasing frequency. We also find the strongest PAC effects in the delta band using the Modulation Index (56) and can thus exclude the possibility that our comparatively new analytic method (OTC) influenced this finding. Nonetheless, we also observed significant coupling to theta and/or alpha phase.

Given this, we trust the delta effects we observe and believe that several factors may account for these results. First, our stimulus presentations occurred in the delta band (with some jitter) and may have therefore partially entrained the visual system. Second, the literature is dominated by PAC findings using phases in a pre-defined frequency band. This is typically theta band activity, in large part motivated by findings in rodents. Recent evidence suggests that the closest analog to rodent theta is in fact human delta oscillations ([62]; Hippocampus; Jacobs et al., 2014, Philosophical Transactions of the Royal Society B). Thus, it is conceivable that previous studies would have also observed strong delta band PAC effects had they considered this band. We note that one strength of the OTC method is that it does not require assumptions about the coupling frequency.

3) How sensitive is detection of HFA event times to the filtering method?

As described above (Reviewer 1, comment 2), we calculated PAC using the Modulation Index and found comparable results. Thus, identifying PAC via detection of HFA events and the exact filtering method do not appear to have an appreciable impact on our primary findings. This comports well with the previous literature (van Vugt et al., 2007, Journal of Neuroscience Methods). We ultimately utilized wavelets based on the OTC method, in which the authors provide an in-depth discussion of the importance of various filtering parameters ([20], Journal of Neuroscience Methods).

4) With regards to electrode choice, the rationale for only using electrodes in the contralateral hemisphere is unclear. Why systematically reject an entire hemisphere (except for 1 subject, oddly) when you visually inspect channels for epileptic activity anyway? Additionally, what is the medical justification for implanting patients with electrodes in what is putatively a healthy hemisphere?

In order to avoid affecting our results by epileptic artifacts, we took an extremely conservative approach by only analyzing electrodes derived from the contralateral hemisphere and subsequently checking these electrodes manually for any remaining epileptic activity. Regarding the medical justification, surgeons typically implant electrodes bilaterally when there is uncertainty regarding the seizure onset zone. In fact, this is a primary reason for performing these implantations and the diagnosis of the affected hemisphere is made well after implantation, often after we have conducted our experiments. The classification of the hemispheres (pathological vs healthy) was based on the results of presurgical diagnostics as well as clinical evaluation of the intracranial recordings. These decisions are made purely based on clinical considerations and none of the authors on this study were involved in this process.

Statistics comments:

1) Watson-Williams test assumes a von Mises distribution. Is this true for the distributions studied here? If not, use the Wheeler-Watson test.

We are in entire agreement with the reviewer. Our implementation of the Watson test used the identical code following Rizzuto et al (2006, Neuroimage) which is publicly available from Mike Kahana’s website. This method explicitly tests the assumption of Von Mises distributions by calculating and comparing the circular dispersion of each distribution. In situations in which the distributions do not share similar dispersions, the test is modified following Fisher (1993).

2) For the resampling statistics: the images were shown in groups of four, but the resampling seems to use random permutation. Resampling should be performed such that the labels for the “chunks” should be shuffled, but within these 4-trial chunks, the labels should be kept the same. This would control for any effect of this chunking.

We apologize for the confusion, which likely arose from our caption for Figure 1. Images were presented in pseudo-random order, not in “chunks”. Our permutation testing is therefore appropriate. We have added text indicating the pseudo-random order of image presentation into the figure caption.

3) Are there still significant differences between categories? How many electrodes have a category with DS=3?

We are unsure what the reviewer means by “still” in this context but 49 of the 63 electrodes showing PAC and phase-clustered HFA events for each of the 4 categories had at least one category with DS=3.

4) It would be nice to also be given an estimate of effect size wherever a p-value is given.

We now include effect sizes for statistically significant (any p<.05) effects which were conducted with binomial, paired t-tests, or chi-square tests. Effect sizes for circular variables appears trickier; after a literature and internet search, we are unaware of effect size calculations for circular data (Rayleigh test, Watson-Williams test). We would be happy to include effect sizes for these tests if the reviewer has further suggestions

5) For the SVM bootstrapping estimates, are the two bootstrapping experiments actually independent in order to support the expected false alarm rate of 0.42 electrodes?

Yes, each of the bootstrapping estimates was performed separately. We either shuffled category labels or used phases at random HFA events. To the extent that shuffling was performed *either* across trials or across HFA events (and never at the same time), we assume that these tests are independent. Nonetheless, our results do show that 17 of the 19 electrodes showing significant decoding compared to random HFA events also shows significant decoding with trial label shuffling. This indicates that an electrode showing significant decoding is likely to show this compared to either of these two types of surrogate testing.

General comments:

1) Are there spatial clusters among the electrodes that have phase coding for each of the different categories (c.f. Vidal et al, 2010)?

We thank the reviewer for this suggestion. We tested this by calculating the average inter-electrode distance between each pairwise combination of electrodes that were selective for a specific category, and compared this to a matched number of randomly drawn electrodes (with replacement). We repeated this randomization procedure 1000 times in order to estimate a distribution of inter-electrode spacings that would be expected by chance. As seen in Figure 6, we neither observed significant spatial clustering nor significantly spatially diffuse electrode configurations (dark blue bars between the 5th and 95th percentile of simulated inter-electrode distances). We believe this lack of clustering is because we were sampling category-selective responses, which extended beyond higher order vision, and have added to text to this effect in the Discussion (see Reviewer 2, comment 1 above for exact text).

Author response image 2.**DOI:**
http://dx.doi.org/10.7554/eLife.07886.021

*2) Please make all rose phase plots opaque as in*
Figure 3
*so that we can see the phase distributions for each category.*

We have updated the figure accordingly.

3) For these phase plots, it would be nice to see the true number of high frequency activity events within each phase bin.

We now include plots showing HFA windows (color coded by the F_MAX_ phase) for each category for these electrodes in the rightmost panels of Figure 3—figure supplement 1 and Figure 3—figure supplement 1. We elected to show these, rather than histograms, because they allow for a better, time-resolved comparison of HFA windows and decoding using power, phase, or HFA (see Reviewer 1, comment 3).

*4) It is unclear how*
Figure 3—figure supplement 1
*should be interpreted. For example, the primary effect in the paper is in the delta range, but this figure seems to show poor delta phase clustering. Why?*

Figure 3—figure supplement 1 shows the proportion of electrodes showing category specific phase-clustering for each category. The largest proportion of electrodes with the most sustained effects is in the delta range at ∼ .5 and 1 Hz (with the exception of Tools). It is also worth noting that since this analysis investigates category specific effects, any significant electrode showing phase clustering will only show up in any one of the 4 panels. Finally, we report phase-clustered HFA events for all 4 categories at F_MAX_ in 63 of 72 PAC+ electrodes. Thus, we observe robust phase-clustering in the delta band which is often category-specific and feel that Figure 3—figure supplement 1 accurately reflects this finding.